# Characteristic Gut Bacteria in High Barley Consuming Japanese Individuals without Hypertension

**DOI:** 10.3390/microorganisms11051246

**Published:** 2023-05-09

**Authors:** Satoko Maruyama, Tsubasa Matsuoka, Koji Hosomi, Jonguk Park, Mao Nishimura, Haruka Murakami, Kana Konishi, Motohiko Miyachi, Hitoshi Kawashima, Kenji Mizuguchi, Toshiki Kobayashi, Tadao Ooka, Zentaro Yamagata, Jun Kunisawa

**Affiliations:** 1Research and Development Department, Hakubaku Co., Ltd., 4629, Nishihanawa, Chuo, Yamanashi 409-3843, Japank.toshiki@hakubaku.co.jp (T.K.); 2Laboratory of Vaccine Materials, Center for Vaccine and Adjuvant Research and Laboratory of Gut Environmental System, Collaborative Research Center for Health and Medicine, National Institutes of Biomedical Innovation, Health, and Nutrition, 7-6-8, Saito-Asagi, Ibaraki 567-0085, Japan; 3Department of Health Sciences, School of Medicine, University of Yamanashi, 1110, Shimokato, Chuo, Yamanashi 409-3898, Japan; 4Artificial Intelligence Center for Health and Biomedical Research, National Institutes of Biomedical Innovation, Health, and Nutrition, 7-6-8, Saito-Asagi, Ibaraki 567-0085, Japan; 5Department of Physical Activity Research, National Institutes of Biomedical Innovation, Health and Nutrition, 1-23-1, Toyama, Shinjuku-ku, Tokyo 162-8636, Japan; 6Institute for Protein Research, Osaka University, 3-2, Yamadaoka, Suita 565-0871, Japan; 7Department of Microbiology and Immunology, Kobe University Graduate School of Medicine, 7-5-1, Kusunoki-cho, Chuo-ku, Kobe 650-0017, Japan; 8Graduate Schools of Medicine, Osaka University, 2-2 Yamadaoka, Suita 565-0871, Japan; 9Graduate School of Pharmaceutical Sciences, Osaka University, 1-6 Yamadaoka, Suita 565-0871, Japan; 10Graduate School of Science, Osaka University, 1-1 Machikaneyamacho, Toyonaka 560-0043, Japan; 11Graduate School of Dentistry, Osaka University, 1-8 Yamadaoka, Suita 565-0871, Japan; 12International Vaccine Design Center, The University of Tokyo, 4-6-1, Shirokanedai, Minato-ku, Tokyo 108-8639, Japan; 13Research Organization for Nano and Life Innovation, Waseda University, 513, Waseda-tsurumaki-cho, Shinjuku-ku, Tokyo 162-0041, Japan

**Keywords:** barley, hypertension, gut bacteria, stratification, machine learning

## Abstract

Background: Barley, a grain rich in soluble dietary fiber β-glucan, is expected to lower blood pressure. Conversely, individual differences in its effects on the host might be an issue, and gut bacterial composition may be a determinant. Methods: Using data from a cross-sectional study, we examined whether the gut bacterial composition could explain the classification of a population with hypertension risks despite their high barley consumption. Participants with high barley intake and no occurrence of hypertension were defined as “responders” (*n* = 26), whereas participants with high barley intake and hypertension risks were defined as “non-responders” (*n* = 39). Results: 16S rRNA gene sequencing revealed that feces from the responders presented higher levels of *Faecalibacterium*, Ruminococcaceae UCG-013, *Lachnospira*, and *Subdoligranulum* and lower levels of *Lachnoclostridium* and *Prevotella* 9 than that from non-responders. We further created a machine-learning responder classification model using random forest based on gut bacteria with an area under the curve value of 0.75 for estimating the effect of barley on the development of hypertension. Conclusions: Our findings establish a link between the gut bacteria characteristics and the predicted control of blood pressure provided by barley intake, thereby providing a framework for the future development of personalized dietary strategies.

## 1. Introduction

Hypertension is a chronic disease with increasing prevalence worldwide, and the most important risk factor for cardiovascular disease (CVD) [1,2]. One in three people suffer from hypertension, and CVD represents one-third of all deaths worldwide [3]. Hypertension seems to be caused by a combination of genetic and lifestyle factors, but according to a large, trans-ethnic, multi-omic study, systolic blood pressure (SBP), diastolic blood pressure (DBP), and pulse pressure explained by gene locus represent less than 5% [4]. In contrast, lifestyle has a substantial impact on hypertension, with high sodium intake showing a direct relationship with the risk of death from CVD [5,6], thus making it the main modifiable target to prevent hypertension.

Recent epidemiological studies have proposed foods that have beneficial effects on hypertension in coronary heart disease [7]. Furthermore, a meta-analysis of 25 randomized, controlled trials reported that dietary fiber intake significantly reduced blood pressure in hypertensive patients [8]. In addition, a clinical study in 25 men and women with mild cholesterolemia reported that barley intake for 5 weeks significantly lowered their DBP [9]. Some recent reports have suggested that gut microbiota is related to the development of hypertension and blood pressure control [10]. For example, a randomized controlled study reported that a fiber-rich oat bran diet reduced blood pressure and improved the β-diversity of gut bacteria [11]. An animal study using spontaneously hypertensive rats reported that elevated blood pressure is associated with dysbiosis of the gut bacteria, a weakening and inflammation of the intestinal epithelial barrier function caused by an imbalance of the microbiome composition [12]. Another study using hypertensive mice reported that a high-fiber diet was accompanied by decreasing blood pressure, an increase in acetate-producing bacteria, and dysbiosis improvement [13]. Furthermore, it was reported that in cancer survivors, the higher the concentration of short-chain fatty acids (SCFAs) in the feces, the lower the blood pressure [14]. These studies suggest that gut bacteria metabolize dietary fiber to produce SCFAs, resulting in lower blood pressure.

Individual differences in the impact on host health are a crucial problem of functional foods [15,16] that has been recently explained, although partially, by the microbiome composition. Barley contains high levels of β-glucan, a soluble dietary fiber, and is a food with potential functions to improve gut bacteria [17,18] and lower postprandial blood glucose levels [19,20]. However, a meta-analysis of 18 randomized control studies suggests that the anti-diabetic effects of β-glucan vary by regions and disease history [21]. In an intervention study conducted by Kovatcheva et al., the examination of the effect of barley on postprandial blood glucose levels showed that some subjects improved their glucose metabolism after barley consumption (responders), while others did not (non-responders). Additionally, gut microbiota measurements revealed that responders had a higher ratio of *Prevotella*/*Bacteroides* compared to that in non-responders [22]. In our previous study, we found that subjects with a high barley intake and no dyslipidemia presented abundant SCFA-producing bacteria, such as *Faecalibacterium* and *Lachnospira*, in their intestines, and we further suggested that individual differences in the barley’s effect on the host lipid metabolism may be due to their intestinal bacterial composition [23]. These findings suggest that the effect of barley on the incidence of hypertension may also differ among individuals due to differences in their gut microbiota composition. Although the mechanisms underlying these effects remain unclear, SCFAs can promote gut barrier integrity, prevent the migration of inflammatory products, and exhibit anti-inflammatory effects via histone deacetylase inhibition and cytokine inhibition [24]. In addition, butyrate may lower blood pressure via afferent vagal signaling [25].

The purpose of this study is to determine whether the functional properties of barley, a food rich in the soluble fiber β-glucan, expected to reduce the risks of hypertension through the production of SCFAs by intestinal bacteria, can be explained by the gut bacterial composition when targeting individual differences in the effect of barley on the risks of hypertension. Here, we defined non-hypertensive subjects with high barley intake as responders, and hypertensive subjects with high barley intake as non-responders, using data from a cross-sectional study in Japanese adults (*n* = 130) aged over 40 years. Comparing the gut bacteria of both groups, we identified some characteristics in the gut bacteria of responders and examined whether these could explain the effects of barley on hypertension. Furthermore, we used machine learning to construct a barley responder classification model based on intestinal bacteria and analyzed whether barley responders could be estimated from their intestinal bacterial composition.

## 2. Materials and Methods

### 2.1. Study Design and Implementation

This is a secondary study to our previous research [23]. This study was based on “the cohort study on barley and the intestinal environment (UMIN000033479)”, an investigation that aimed to evaluate the association between barley and intestinal bacteria among employees at a company that manufactures barley products. The original study was conducted in accordance with the principles of the Declaration of Helsinki, and sampling was performed between August 2018 and March 2019. The detailed study design and methods of implementation have been previously reported [23]. Briefly, among the 272 individuals who provided informed consent, we selected 130 participants aged 40 or older, as they were considered to be at high risk due to lifestyle-related diseases. All participants were asked to answer a questionnaire related to their physical activity and smoking habits (never, current, past), and provide a copy of their medical checkup to obtain basic information, such as weight and blood tests results. In addition, we assessed the participants’ total daily energy intake (kcal/d) through a brief-type, self-administered dietary history questionnaire (BDHQ, Gender Medical Research, Inc., Tokyo, Japan). The daily barley intake (g/d) was determined by the bowl size (200 g for large, 160 g for medium, 140 g for small, and 100 g for kids’ size), the percentage of barley mixed with white rice (none, 5, 10, 15, 30, or 50%), the number of intakes per day (times), and the frequency of consumption per month (none, 0.5, 1, 4, 8, 16, or more d/month). Then, we calculated the daily barley consumption (g/1000 kcal) from the total daily energy (kcal/d) and barley intake (g/d). The 130 test subjects were subsequently classified into either a “high barley group” (*n* = 65) or a “low barley group” (*n* = 65), based on their median barley intake (3.68 g/1000 kcal). The primary analysis group for this study was the high barley group. The high barley group was stratified into responders or non-responders depending on whether they were at risk for hypertension. Subjects were categorized as non-responders if they met at least one of the following: (1) SBP ≥ 130 mmHg, (2) DBP ≥ 85 mmHg, or (3) they were receiving treatment and were considered to be at risk for developing hypertension. Thus, non-responders included individuals with border zone hypertension (SBP 130–140 mmHg or DBP 85–90 mmHg). Figure 1 shows the flowchart for the selection of participants in this study.

### 2.2. DNA Extraction and 16S rRNA Gene Amplicon Sequencing

We requested participants to collect fecal samples at home in containers with a guanidine thiocyanate (GuSCN) solution (TechnoSuruga Laboratory, Shizuoka, Japan). These samples were stored at room temperature and DNA was extracted within 5 days following previous methods [26]. Briefly, fecal samples in 0.2 mL of GuSCN solution were homogenized with 0.3 mL of lysis buffer (No. 10, Kurabo Industries Ltd., Osaka, Japan), 0.5 g of 0.1 mm glass beads (WakenBtech Co., Ltd., Tokyo, Japan), and 0.2 mL of GuSCN solution using a Cell Destroyer PS1000 (Bio-Medical Science, Tokyo, Japan) at 4260 rpm for 50 s at room temperature. Then, the sample was centrifuged at 13,000× *g* for 5 min at room temperature, and DNA was extracted from the supernatant solution using the Gene Prep Star PI-80X instrument (Kurabo Industries, Osaka, Japan). DNA concentration was measured using a NanoDrop Spectrophotometer ND-1000 (Thermo Fisher Scientific, Waltham, MA, USA), and extracted DNA was stored at −30 °C. Amplicon sequencing was performed using the 16S rRNA V3–V4 region, as previously reported [26]. Barcoded amplicons were generated by the following primers: forward, 5′-TCGTCGCAGCAGCAGATGTGTAGTAAGCGACAGCCTACGGNGGCWGCAG-3′; reverse, 5′-GTCT CGTGGCTCGAGATGTATAAGACGACTACHVGGTATCTAATCC-3′. We used the Illumina MiSeq instrument (Nextera XT Index Kit v2 Set A, Illumina, San Diego, CA, USA) to construct DNA libraries. Library concentrations were determined using a QuantiFluor dsDNA System (Promega Co., Madison, WI, USA), and 16S rRNA gene sequencing was performed using the Illumina MiSeq instrument (Illumina). In cases without detailed methods, we followed the company’s recommendations.

### 2.3. Bioinformatics Analysis

We used the Quantitative Insights Into Microbial Ecology (QIIME) software package (v. 1.9.1) to analyze the sequences [27]. Operations from paired-end reads trimming to operational taxonomic unit (OTU) picking were performed automatically using the QIIME Analysis Automating Script (Auto-q) [27]. OTUs were selected based on sequence similarity (>97%) using the open-reference OTU picking with UCLUST software against the SILVA v128 reference sequences, and the taxonomy (phylum, class, order, family, genus) and relative abundance were calculated [28,29]. Ten thousand reads were randomly selected for the statistical analysis.

### 2.4. Statistical Analysis

All statistical analyses were performed using R version 3.6.0 unless otherwise indicated [30].

#### 2.4.1. Calculation of α-Diversity and Intergroup Comparison of Subjects

The α-diversity (within-subject bacteria diversity) indexes Chao1, Shannon, and Simpson were calculated using the richness function from the phyloseq R-package. Age, body mass index (BMI), blood pressure, fasting blood glucose, hemoglobin (Hb)A1c, triglycerides, low-density lipoprotein (LDL)-cholesterol, high-density lipoprotein (HDL)-cholesterol, and dietary intake (per 1000 kcal) between the responders and the non-responders were compared employing Student’s *t*-test. We performed the Mann–Whitney *U*-test to compare the α-diversity and the abundance of gut bacteria between the responders and the non-responders. The gut bacteria were compared based on the top 50 genera, which were sorted by mean relative abundance in the study subjects. The significance level for all statistical analyses was set at *p* < 0.05.

#### 2.4.2. Comparative Analysis Excluding Participants Undergoing Treatment

Since the gut microbiome of the treated subjects in the non-responders could influence the outcome, we statistically compared the gut bacteria of responders and non-treated non-responders while excluding participants who were taking therapeutics (calcium channel blocker, angiotensin II receptor blocker, any other antihypertensive drug, antidiabetic, or antihyperlipidemic) using the Mann–Whitney *U*-test.

#### 2.4.3. Principal Coordinate Analysis

This analysis was performed by combining the two-dimensional data of gut bacteria at all genus levels through Principal Coordinate Analysis (PCoA) using the vegdist and quasieyelid functions from the vegan R-package and the dudi.pco function from the ade4 R-package. Data were calculated using the Bray–Curtis distance. All participants (*n* = 130) were included in this analysis, and color-coded plotted as low barley group (*n* = 65), non-responders (*n* = 26), and responders (*n* = 39). PCo1 and PCo2 between groups were compared utilizing the Mann–Whitney *U*-test.

#### 2.4.4. Logistic Regression Analysis

We analyzed the relationship between the responder and their characteristic bacteria using logistic regression analysis to minimize the effects of confounding factors. We included the responders (1 = responder, 0 = non-responder) as the objective variable, and each bacteria and other adjusting covariates (sex, age, BMI, fasting blood glucose, triglycerides, LDL-cholesterol, smoking, physical activity, and parents with hypertension) as the explanatory variable. We evaluated the variance inflation factors (VIFs) using the vif function of the car R-package. All VIFs were less than 2 and were accepted for inclusion in the model.

#### 2.4.5. Random Forest Machine Learning

We developed a classification model based on the random forest supervised learning algorithm to estimate the responders. As the variables for the model, we set the top 20 (with an overall average of 1% or more), top 50, top 64 (with an overall average of 0.1% or more), and top 100 genera found in the 130 subjects. The gut bacteria data from 47 randomly selected subjects (70% of the high barley group) served as the training set, and the gut bacteria data from the remaining 18 subjects served as the test set. The classification model was generated using the RandomForest and the caret R-packages. We performed a repeated cross-validation to cope with the small sample size and to improve the model performance, setting the number of cross-validation folds to 13 and repeating it 10 times. Hyperparameter ntree was set to 500, and mtry was tuned using the caret R-package. Other parameters were set to the default values suggested by the RandomForest R-package. We performed 200 iterations each for the model. The importance of each variable was calculated using the varImp function from the caret R-package. We ran the model using the receiver operating characteristic (ROC) curve (ROCR R-package) and calculated the area under the ROC curve (AUC) to evaluate the model performance.

## 3. Results

### 3.1. Subject Characteristics and Barley Responder/Non-Responder Definitions

Table 1 shows the background data for the responders and the non-responders in the high barley group (*n* = 65). There was no significant difference in barley intake between responders and non-responders. Further, barley intake duration was confirmed by categorical questions (over 10 years, 5–10 years, 3–5 years, 2–3 years, 1–2 years, 6–12 months, 3–6 months, or less than 3 months) and no significant difference between responders and non-responders was observed. Owing to this result, we did not consider the duration of the barley intake in further analyses (Appendix A). Moreover, age, weight, BMI, fasting blood glucose, and LDL-cholesterol were found to be higher in the non-responders. Their mean fasting blood glucose (100 mg/dL) was normal/slightly high and mean LDL-cholesterol (133 mg/dL) was diagnosed as borderline hypercholesterolemia; thus, the non-responders were classified as presenting pre-metabolic syndrome based on medical factors other than blood pressure, and were considered a high-risk population for diabetes and dyslipidemia in addition to hypertension. Furthermore, HbA1c was 5.5 (SD = 0.3) for the responders and 5.7 (0.5) for the non-responders, indicating that many individuals in the high barley group presented prediabetes. In addition, previous studies have reported that no differences existed in the blood pressure between the low and high barley groups [19]. When probed, seven of the twenty-six non-responders were on hypertension medications, and most of them were calcium channel blockers (*n* = 7) and angiotensin II receptor blockers (*n* = 1). According to the previously published report and our results, it is likely that the non-responders had parents with hypertension.

The results from the dietary survey showed that salt intake was 9.6 ± 2.4 g/d in the barley responders and 10.1 ± 3.1 g/d in the non-responders; thus, there was no difference between the two groups (*p* = 0.49, Student’s *t*-test), and these values were similar to the average salt intake reported by the National Health and Nutrition Survey Japan 2018 [31]. Although oral salt intake is known to be a direct factor influencing elevated blood pressure [32,33], our stratification of barley responders and non-responders was unaffected by salt intake. Furthermore, there were no significant differences between the groups in smoking and physical activity, which are two important lifestyle factors that can affect blood pressure. Therefore, the individuals selected for this study were suitable for observing the association between barley and intestinal bacteria.

Background data for the low-barley group are presented in Appendix A. Significant differences between the low and high-risk groups for hypertension in the low barley group were also observed for several indices other than blood pressure (Appendix A).

### 3.2. Intestinal Bacteria Characteristics of Barley Responders

To investigate the gut bacteria associated with the classification of high barley intake as a risk of hypertension, we compared the microbiome in the barley responders and that of the non-responders. Regarding α-diversity, the responders showed a significantly lower Chao1 index (*p* = 0.009), while the Shannon index tended to be higher (*p* = 0.07) (Table 2). Subsequently, we compared the gut microbiota of all subjects using PCoA and observed a significant difference between the responders and the non-responders in PCo2 (*p* = 0.008) (Figure 2). There was also a significant difference between the responders and the low barley group in PCo2 (*p* = 0.006), and no difference between the non-responders and the low barley group (*p* = 0.50) (Figure 2). This indicated that the microbiome of the responders was specific. We next compared the gut bacterial composition of the responders and the non-responders. Among the top 50 genera (>0.213%) in terms of relative abundance, *Faecalibacterium* (*p* = 0.02), *Lachnospira* (*p* = 0.02), Ruminococcaceae UCG-013 (*p* = 0.03), and *Subdoligranulum* (*p* = 0.04) were significantly higher in the responders, and *Prevotella* 9 (*p* = 0.03) and *Lachnoclostridium* (*p* = 0.02) were significantly lower in this group (Table 2). Thus, the barley responders presented a characteristic gut bacterial composition, suggesting that the physiological effects of these gut bacteria were associated with the maintenance of blood pressure homeostasis in the host. We confirmed a similar trend in a stratified analysis that excluded non-responders who were using therapeutic medicines (Appendix A).

### 3.3. Adjusted Relationship between Responders and Characteristic Bacteria

As shown in Table 1, some differences were observed in background factors between responders and non-responders. In particular, seven (27%) of the non-responders were already taking medications for hypertension, which may have altered the composition of their gut microbiota, resulting in changes between non-responders and responders. Therefore, we compared the intestinal bacteria of the non-responders (*n* = 19), excluding the seven participants who were taking hypertension medications, and the responders (*n* = 39) using the Mann–Whitney *U*-test. The results showed that the six genera of intestinal bacteria characteristic of responders remained significantly different between the two groups (Appendix A).

In addition, parents’ history of hypertension may have been an important confounding factor; therefore, we adjusted for it in the logistic regression analysis. Adjusted factors for this model were sex, age, lifestyle-related indicators, and parents’ history of hypertension. The results showed that the association between responders and characteristic gut bacteria was weakened by parents’ history of hypertension. However, Ruminococcaceae UCG-013 remained positively associated with the responders (*p* = 0.04), while *Faecalibacterium* showed an insignificant but positive association (*p* = 0.06) (Appendix A).

Next, we conducted a logistic regression analysis to analyze whether there was an association between responders and characteristic bacteria after adjusting for sex, age, lifestyle-related diseases, and lifestyle habits. A positive association with responders was found for *Faecalibacterium* (*p* = 0.015) and Ruminococcaceae UCG-013 (*p* = 0.04), and a negative association for *Lachnoclostridium* (*p* = 0.04). Moreover, *Lachnospira* (*p* = 0.099), *Prevotella* 9 (*p* = 0.097), and *Subdoligranulum* (*p* = 0.07) showed a non-significant association (Appendix A).

### 3.4. Determination Model Based on the Gut Bacteria of Barley Responders

We observed a marked difference in bacterial composition between the barley responders and non-responders. Therefore, we attempted to estimate the barley responders by their intestinal bacterial composition using a random forest model. The AUC of the classification model was improved by repeated cross-validation.

The gut bacteria used as explanatory variables were the top 20 genera (mean abundance > 1%), 50, 64 (mean abundance > 0.1%), or 100 genera. Based on each explanatory variable, 200 iterations were conducted for the model. The results showed that the mean (SD) AUC of the test data was 0.58 (0.06), 0.65 (0.03), 0.73 (0.02), and 0.59 (0.03) for explanatory variables 20, 50, 64, and 100 genera, respectively. The most accurate model included 64 explanatory variables, with an optimal cutoff of 0.562, sensitivity of 0.82, specificity of 0.71, and overall accuracy of 77%. The AUC for the test set (the final model rating) was 0.805 (Figure 3a); as a reference, the AUC for the training set was 1.0. The top 20 gut bacteria essential to explain this classification model included *Lachnoclostridium*, *Prevotella* 9, *Lachnospira*, *Subdoligranulum*, Ruminococcaceae UCG-013, and *Faecalibacterium* (Figure 3b), which were generally consistent with the bacterial characteristics of the responders listed in Table 2. Thus, our model reflected the taxonomic properties characteristic of the responders. Appendix A shows representative ROC curves for different gut bacteria included as explanatory variables.

## 4. Discussion

To the best of our knowledge, this is the first study to relate individual differences in the effects of barley on blood pressure to gut bacteria. In general, the consumption of fiber-rich foods is associated with improved blood pressure [9,34], but the effect varies among individuals. This study highlights a potential mechanism that is not the direct lowering of blood pressure by barley, but rather is associated with the clear differences in the function of gut bacteria. The potential role of the gut microbiota on blood pressure has received considerable attention in recent decades [34,35,36], and the roles of SCFAs, especially propionate, were reported to be among the significant agents for inhibiting angiotensin [34]. Moreover, a previous study on animals reported that a fecal microbiota transplant from hypertensive mice into germ-free mice resulted in the animals developing hypertension [37]. In addition, several studies have reported an association between hypertension and dysbiosis, indicating that the intestinal bacterial equilibrium plays an important role in the development of hypertension and in blood pressure control [38,39]. Consistent with the findings of these studies, the non-responders in this study tended to have a lower Shannon index, which was calculated based on the proportion of species in a gut microbiome. This group may be depleted regarding the generally dominant gut bacteria or may have an imbalanced intestinal bacterial composition and associated inflammation of the intestinal epithelium. Conversely, non-responders had a significantly higher Chao1 index, calculated by weighting rare species of bacteria. Furthermore, the PCo2 component of the microbiome of the responders was also significantly lower than that of the low barley group (Figure 2), indicating that the responders had a characteristic microbiome.

To identify the gut bacteria specific to the responders, we next performed a comparative analysis based on the top 50 genera sorted by their mean relative abundance of 130 subjects. The results showed that the responders were depleted of *Prevotella* 9 and *Lachnoclostridium*, and were abundant in *Faecalibacterium*, *Subdoligranulum*, *Lachnospira*, and Ruminococcaceae UCG-013, compared to the non-responders. These six genera may contribute to the classification of risks of hypertension by barley, as there were no differences observed between the hypertensive and non-hypertensive individuals in the low barley group (Appendix A). Although the results of the cross-sectional study revealed that these six genera were responder-specific gut bacteria, we were unable to further probe and specify whether these six genera were increased particularly in responders due to barley consumption, or were originally more abundant.

A previous study reported an increased abundance of *Prevotella* 9 in hypertensive patients and suggested that the appearance of inflammation due to increased *Prevotella* 9 was associated with increased blood pressure [40]. In fact, *Prevotella copri*, which belongs to the same group as *Prevotella* 9, is a gut bacterium strongly associated with rheumatoid arthritis and has been reported to induce inflammation [41,42]. It has also been reported that the superoxide reductase and phosphoadenosine phosphosulphate reductase encoded by *P. copri* are associated with the promotion of inflammation and exacerbate the inflammation of the intestinal epithelium in colitis model mice [43]. Conversely, while *P. copri* is known to cause intestinal inflammation, the interaction of *Prevotella* and dietary fiber has frequently been reported to have a positive impact on the host’s health [44,45]. For example, the consumption of cereal fiber for 8 weeks improved CVD markers accompanied by an increase in *P. copri* in a randomized, controlled trial in 47 individuals [44]. A crossover, randomized, controlled trial in 19 mildly hypercholesterolemic individuals taking barley β-glucan for 35 days also found a tendency to increase in *Prevotella,* accompanied by a negative correlation between *Prevotella* and triglycerides [45]. Thus, the effect of barley consumption on changes in the *Prevotella* 9 abundance in the gut of hypertensive individuals remains unclear, but the finding that *Prevotella* 9 was depleted in the intestines of responders in this study could be important in discussing the association between barley and intestinal inflammation. Further studies are needed to determine how barley consumption affects the *Prevotella* 9 abundance in the gut and the associated immunological changes. In summary, these results indicate that responders who consumed barley and did not have hypertension exhibited a higher gut bacterial diversity and Shannon index, and may aid in preventing dysbiosis. Additionally, barley intake did not specifically correlate to the levels of inflammation-causing gut bacteria such as *Prevotella* 9.

Changes in the gut bacterial composition are also thought to affect the host by altering the production levels of microbial metabolites, specifically SCFAs derived from dietary fiber fermentation. SCFAs are an important source of nutrition for intestinal epithelial cells and are beneficial for the regulation of growth, differentiation, and the function of intestinal secretory and other types of cells; additionally, they are considered a potential factor in blood pressure homeostasis [46]. Furthermore, it has been reported that SCFAs promote blood pressure reduction via the upregulation of the host G protein-coupled receptors Gpr41 and Olfr78 in a mechanism independent of salt intake [47,48]. In our study, the dietary survey results showed no differences in salt intake between the responders and the non-responders. Thus, our findings indicate that the abundance of characteristic gut bacteria in the responders is not attributed to the differences in salt intake. In contrast, the reverse causality may have prevented the high salt intake in non-responders, and therefore a prospective observation is needed. Four genera of enterobacteria that were enriched in the responders are known to consistently produce SCFAs. A previous study has revealed that the avoidance of hypertension is at least in part associated with the abundance of SCFAs-producing bacteria [37], and the abundance of these bacteria may be related to the classification of responders in the high barley group. *Faecalibacterium*, *Subdoligranulum*, and Ruminococcaceae UCG-013 belong to the Ruminococcaceae family, and many bacteria in this family are known to be common producers of butyrate. Although the details of the Ruminococcaceae UCG-013 butyrate metabolic pathway are unknown, there are many strains of *Faecalibacterium* and *Subdoligranulum* with well-defined pathways for butyrate production. Furthermore, several studies have reported that *Faecalibacterium* is depleted in hypertensive patients, suggesting that the metabolites produced by these bacteria are closely related to blood pressure control [40,49]. *Faecalibacterium prausnitzii*, the only known species belonging to *Faecalibacterium*, is known to degrade dietary fiber [50,51] and is expected to convert barley β-glucan into SCFAs. In addition, the main source for *F. prausnitzii* butyrate production is attributed to the cross-feeding of acetate produced by other bacteria [52], suggesting that it produces butyrate via interactions with other barley-utilizing bacteria. However, in this study, no correlation was found between *Faecalibacterium* and other well-known acetate-producing bacteria (e.g., *Bifidobacterium*). The biological activity of *Subdoligranulum variabile*, the only known species belonging to *Subdoligranulum*, is largely unknown; however, the difference in the 16S rRNA sequence between *S. variabile* and its most closely related species, *F. prausnitzii*, was 9% [53], and the two species were expected to have similar functions. In other words, our finding that *Faecalibacterium* and *Subdoligranulum* were enriched in the responders in this study is likely related to the fermentation of barley β-glucan and the conversion of the acetate supplied by the other bacteria into butyrate. *Lachnospira* is a well-known SCFA producer [54] whose intestinal abundance has also been reported to positively correlate with dietary fiber intake [50]. An intervention study in a healthy individual reported a positive correlation between *Lachnospira* and acetate and butyrate in feces when whole grains were consumed [55], suggesting that barley may be metabolizable. On the other hand, gut microbiota studies in patients with irritable bowel syndrome and Crohn’s disease have not yielded consistent results on the abundance of *Lachnospira* [56,57,58], and further studies are needed to clarify how the immunological changes in the gut caused by *Lachnospira* may affect hypertension. However, as all the gut bacteria enriched in the responders in this study were SCFA-producing bacteria, it was inferred that individuals presenting an abundance of these bacteria may be able to benefit from barley to improve their blood pressure control. Additionally, although γ-aminobutyric acid (GABA) is known to be one of the beneficial metabolites of intestinal bacteria that is well-known to prevent hypertension, bacteria belonging to *Bacteroides*, *Parabacteroides*, and *Escherichia*, which have been reported to belong to GABA-producing genera [59], showed no differences between responders and non-responders.

Although many common oral medications affect gut bacterial composition [60], we confirmed that the gut bacteria of non-responders remained unaffected by medications for hypertension (Appendix A). However, the sample may have been too small to examine the effect of hypertension medications. A longitudinal study is needed to determine how the effect of hypertension medications on gut bacteria is related to barley intake. Furthermore, the genetic influence on hypertension cannot be ignored. In fact, 85% of non-responders had a parental history of hypertension (Table 1). Therefore, parental hypertensive history may be a confounding factor and was adjusted by logistic regression analysis. As a result, it was found that Ruminococcaceae UCG-013 was significantly associated with responders (*p* = 0.04) even after excluding confounding factors, and *Faecalibacterium* also tended to be associated (*p* = 0.06). These two genera were therefore identified as gut bacteria that particularly underscore the relationship between barley intake and responders. Additionally, since it is well-known that hypertension tends to cluster with other risk factors such as dyslipidemia, insulin resistance, and obesity [61], which may act as confounding factors, we examined the impacts of these confounders using logistic regression. Our results revealed the significant effects of *Faecalibacterium*, *Lachnoclostridium*, and Ruminococcaceae UCG-013 on the classification of responders even after removing the effects of confounding factors (Appendix A). Moreover, the other three bacteria (*Lachnospira*, *Prevotella* 9, and *Subdoligranulum*) exhibited an acting tendency on the responder’s classification (Appendix A). Thus, these risk factors were not confounded enough to influence our conclusion; these results implied that the characteristic gut bacteria of responders were partially associated with blood pressure.

In an additional analysis, we established a responder classification model based on gut bacteria. In epidemiological studies, individual differences in patients are usually identified as confounding factors and excluded, e.g., by multivariate analysis. However, in this study, we focused on these individual differences and were able to show that gut bacteria can partially explain differences in the effects of barley. This classification model can help explain host responses to nutrition in the future. Additionally, *Bifidobacterium*, *Blautia*, and *Butyricicoccus* have been reported to increase with barley consumption [17,18]; in this study, *Blautia* and *Butyricicoccus* were found to be highly important gut bacteria in the responder prediction model (Figure 3). Therefore, it was suggested that individuals enriched with these gut bacteria could be classified in terms of their barley consumption. Furthermore, in the low barley group, these bacteria were not associated with the risks of hypertension, suggesting that barley consumption may have increased these bacteria in responders. However, to determine how continuous barley intake affects the abundance of these bacteria, a longitudinal study is needed.

This study has several limitations. Firstly, all the participants were employees at a company that manufactures barley products, and they consumed barley more frequently than the average Japanese individual. Furthermore, the possibility that they may have exaggerated their barley intake to positively impress the company cannot be ruled out. We will need to examine the accuracy of the questionnaire used to estimate barley intake in the future. It is also difficult to generalize data sets including widely different populations (i.e., race and dietary habits), which may have affected the model performance. Secondly, we did not analyze the metabolites and function of the bacteria. Therefore, we could not refer to the details or mechanism of our results. Thirdly, in this study we used a 97% OTU method which has recently shifted to an amplicon sequence variants (ASVs) method. This might influence the result of the α-diversity, especially the Chao1 index, because the Chao1 index weighs rare bacteria. On the other hand, it might not influence the mainstream of this study because we used only 50 dominant bacteria to compare them in the responders and non-responders. Finally, this was a cross-sectional study, and we could not prove causality; hence, we only discussed the associations. Future studies should test the performance of our classification model in unbiased subjects and expand it to other populations with different races and lifestyles to identify potential confounding factors. We are further planning to attempt a long-term, longitudinal investigation of the effect of the gut bacterial characteristics of the responders in improving blood pressure control. While this study defined the differences between the responders and the non-responders by their intestinal bacteria, in the future we would examine how a non-responder can transition into a responder. If everyone could experience the benefits of barley, it could help reduce the risk of CVD worldwide.

## 5. Conclusions

In this study, we identified the gut bacteria that are characteristic of barley responders of hypertension. The hypertensive non-responder was defined as a person who consumes barley but has blood pressure levels above a border zone or is receiving treatment for hypertension. The possibility of stratifying the host’s blood pressure homeostasis due to barley intake presented in this study provides new knowledge to solve the individual differences in efficacy that have been an issue in functional studies on various foods. In particular, the machine-learning responder classification model created for this study could determine the compatibility between barley and the host based on their intestinal bacteria. We hope this study will provide important insights into designing personalized dietary treatments to prevent CVD in the future.

## Figures and Tables

**Figure 1 microorganisms-11-01246-f001:**
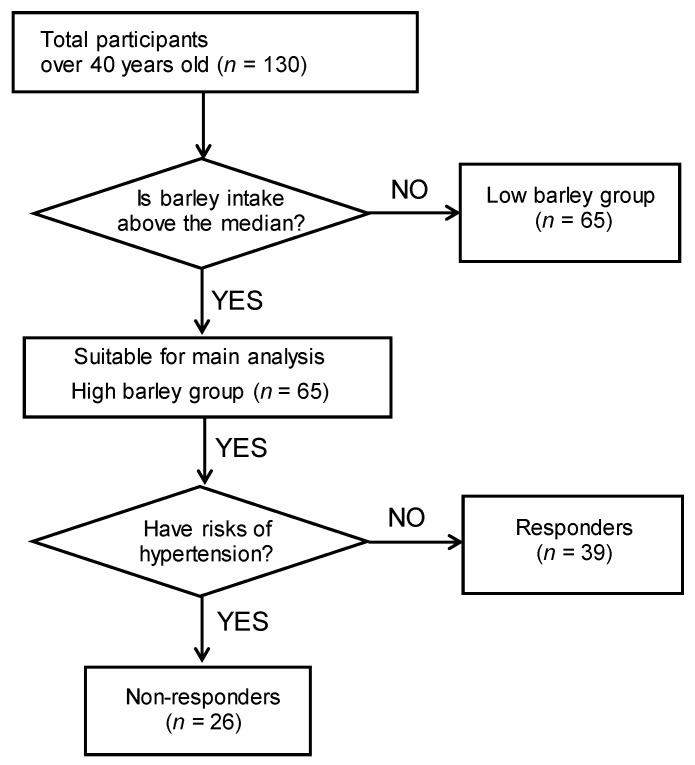
Flowchart of the recruitment and selection of participants for this study.

**Figure 2 microorganisms-11-01246-f002:**
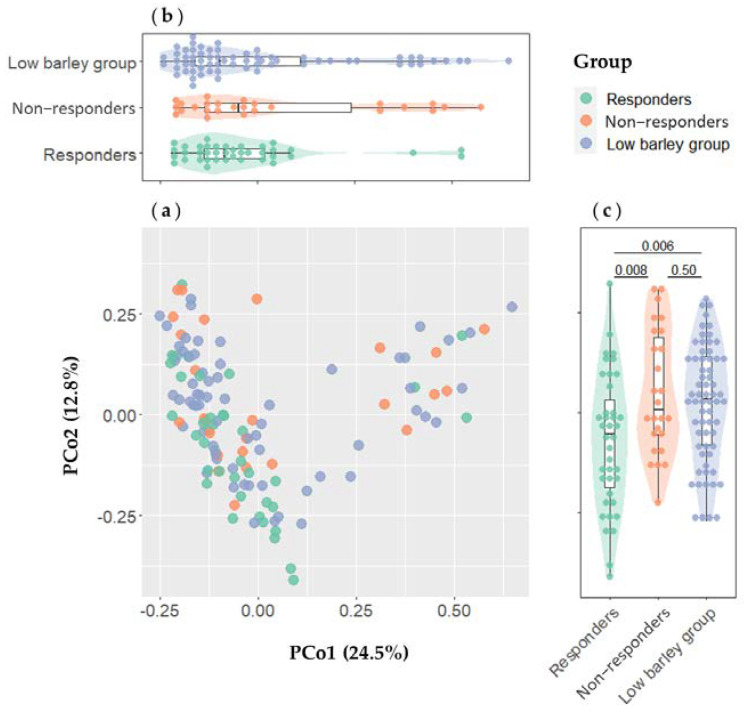
Comparison of the gut microbiome composition in the responders (*n* = 39), non-responders (*n* = 26), and low barley group (*n* = 65). (**a**) Scatter plots of PCo1 and PCo2. (**b**) Comparison of PCo1. (**c**) Comparison of PCo2. PCoA of the gut microbiome was based on an abundance of 266 genera. PCo1 and PCo2 between groups were compared using the Mann–Whitney *U*-test.

**Figure 3 microorganisms-11-01246-f003:**
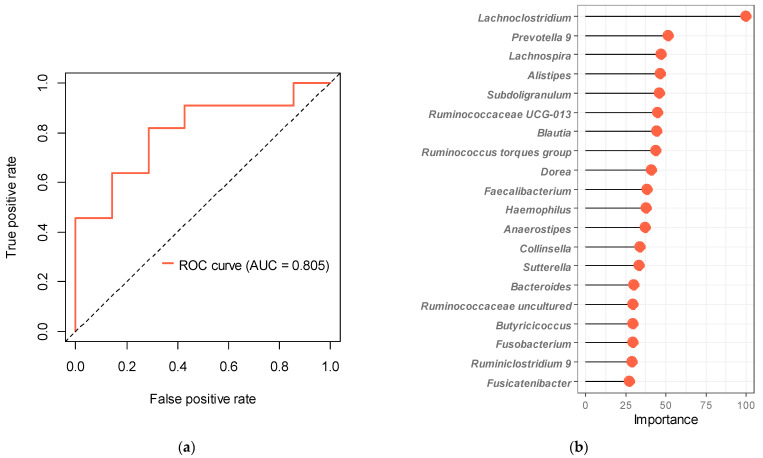
Random forest classification model generated based on 64 genera included in the training data set. (**a**) ROC curve and AUC of the microbiome-based model for the discrimination between the responders and the non-responders. (**b**) Top 20 genera of intestinal bacteria (explanatory variables), sorted by importance. Dots show the importance of each intestinal bacterium.

**Table 1 microorganisms-11-01246-t001:** Characteristics of medical checkup and dietary habits in both study groups.

	Non-Responders (*n* = 26)	Responders (*n* = 39)	*p* Value ^(1)^
	*n* (%) or Mean (SD)	*n* (%) or Mean (SD)	
Male (*n*)	24 (92%)	28 (72%)	0.09 ^(2)^
Age (years)	53.5 (6)	48 (6)	0.002
Barley intake (g/1000 kcal)	10.5 (5.7)	8.4 (3.4)	0.11
Medications of hypertension drug (*n*)	7 (27%)	0 (0%)	<0.001 ^(2)^
Parents with hypertension (*n*)	21 (81%)	15 (38%)	0.002 ^(2)^
Medical checkup			
Weight (kg)	71.9 (9.7)	65.6 (13.4)	0.03
BMI (kg/m^2^)	24.9 (3.4)	22.8 (3.4)	0.02
SBP (mmHg)	138 (16)	113 (10)	<0.001
DBP (mmHg)	92 (8)	73 (9)	<0.001
Fasting blood glucose (mg/dL)	100 (17)	89 (7)	0.004
HbA1c (%)	5.7 (0.5)	5.5 (0.3)	0.15
Triglycerides (mg/dL)	149 (75)	111 (92)	0.07
HDL-cholesterol (mg/dL)	57 (15)	61 (19)	0.27
LDL-cholesterol (mg/dL)	133 (27)	115 (29)	0.01
Nutrients			
Energy (kcal/d)	1815 (569)	1746 (439)	0.60
Protein (g/d)	60 (22)	59 (17)	0.90
Fat (g/d)	48 (20)	52 (18)	0.34
Carbohydrate (g/d)	236 (99)	224 (68)	0.60
Sodium chloride (g/d)	10.1 (3.1)	9.6 (2.4)	0.49
Lifestyle			
Smoking (present, past, never) (*n*)	5, 14, 7 (19%, 54%, 27%)	12, 12, 15 (31%, 31%, 39%)	0.18 ^(2)^
With physical activity ^(3)^ (*n*)	4 (23%)	22 (85%)	0.66 ^(2)^

^(1)^ Responders and non-responders were compared using Student’s *t*-test. ^(2)^ Responders and non-responders were compared using Pearson’s chi-squared test. ^(3)^ Light exercise for a total of at least 30 min/day, at least twice a week, for at least 1 year.

**Table 2 microorganisms-11-01246-t002:** Characteristics of the microbiome in both study groups.

	Non-Responders (*n* = 26)	Responders (*n* = 39)	*p* Value ^(1)^
	Median [Interquartile Range]	Median [Interquartile Range]	
α-Diversity			
Chao1	1028 [806, 1331]	997 [895, 1210]	0.009
Shannon	3.47 [3.22, 3.80]	3.66 [3.32, 3.91]	0.07
Simpson	0.92 [0.89, 0.95]	0.94 [0.90, 0.95]	0.20
Genus			
*Faecalibacterium*	1.74 [0.54, 5.28]	5.86 [2.09, 9.28]	0.02
*Lachnoclostridium*	2.12 [0.93, 2.84]	1.13 [0.71, 1.75]	0.02
Ruminococcaceae UCG-013	0.08 [0.003, 0.19]	0.21 [0.08, 0.50]	0.03
*Lachnospira*	0.15 [0.003, 0.52]	0.62 [0.22, 1.37]	0.02
*Prevotella* 9	0.01 [0.00, 24.19]	0.00 [0.00, 0.01]	0.03
*Subdoligranulum*	0.26 [0.003, 1.39]	1.69 [0.09, 2.58]	0.04

^(1)^ Responders and non-responders were compared using Mann–Whitney’s *U*-test.

## Data Availability

DNA sequencing data generated in this study have been deposited in the DNA Databank of Japan (DDBJ) under the accession number DRA016125. Other data and analysis codes described in this manuscript are available from the corresponding author upon request.

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
