# Peer review of "Characteristic Gut Bacteria in High Barley Consuming Japanese Individuals without Hypertension"

_microorganisms, 2023, doi:10.3390/microorganisms11051246_

Round 1

Reviewer 1 Report

Maruyama et al attempted to investigate if the gut microbial composition could explain the difference between high-barley-intake adults with and without hypertension. To achieve this, they leveraged a cohort with 26 responders (participants with high barley intake and no hypertension) and 39 non-responders (individuals with high barley intake and hypertension risks). They compared the microbial compositions of responders to those of non-responders and found some taxa that display differences in microbial relative abundances such as Faecalibacterium. Finally, they designed a random forest model to distinguish two groups based on their microbial compositions, reporting a decent performance of AUROC=0.75. Based on their classification model, they found some important taxa that coincide with their initial correlation analyses. I am quite open to looking at a revised version if the authors could address some major and minor issues in a satisfactory fashion, which we describe in more detail below.

Major issues:

1.     I worry about the lack of control. More specifically, I think it would be better if the authors have other groups of participants such as (1) people with low barley intake and no hypertension and (2) people with low barley intake and hypertension. In other words, I worry about the fact that this study only focused on participants with high barley intake. I think all available groups shall be used in the random forest model for the classification.

2.     In the method description of the random forest model (lines 207-208), they only used the top 50 genera found in the 130 subjects. What is the reason why only the top 50 genera were kept? Is this because the best classification performance can be achieved when the top 50 genera were used? I think the number of genera included should be considered as a hyperparameter that needs to be tuned based on the cross-validation results of the training set.

3.     I think a more detailed discussion over why and how the gut microbes impact the metabolism is lacking on lines 72-75 and lines 86-90. Does dietary intake such as barley change the gut microbiome then the gut microbiome changes the metabolism (Tong Wang et al., PloS Computational Biology 2019)?

Minor comments:

1.     Line 288: “Subdoligranulum (p = 0.07) showed non-significant trended association…” -> “Subdoligranulum (p = 0.07) showed a non-significant association…”

2.     Line 341: “responders specific gut bacteria…”  -> “responder-specific gut bacteria…”

3.     Line 367: “…levels of inflammation causing gut bacteria such as Prevotella 9.” -> “…levels of inflammation-causing gut bacteria such as Prevotella 9.”

4.     Line 404: “Lachnospira is a well-known SCFA-producer…” -> “Lachnospira is a well-known SCFA producer…”

5.     Line 444: “There are several limitations in this study.” -> “This study has several limitations.”

6.     Line 464: “it could help reducing the risk…” -> “it could help reduce the risk…”

7.     Line 475: “We hope this study will provide important insights to designing personalized…” --> “We hope this study will provide important insights into designing personalized…”

Reviewer 2 Report

In this manuscript, Satoko Maruyama et al. describe a study of characteristic gut bacteria in high barley consuming Japanese 2 individuals without hypertension. Barley is expected to lower blood pressure. And the composition of gut microbiota may affect the regulatory effect of barley on hypertension. The overall subject is meaningful and worthy of study. However, this article has serious flaw.

Table 1, page7: Barley intake showed no significant difference between Non responders and Responders, while Medications of hypertension, Parents with hypertension, Weight, and BMI showed significant differences between Non responders and Responders. Studies have shown that Medications of hypertension, Parents with hypertension, Weight, and BMI may all have an impact on the structure of the gut microbiota. Therefore, the differences in gut microbiota found in this study may not be caused by different intake of Barley, but rather by Medications of hypertension, Parents with hypertension, Weight, and BMI.

Are all 16S raw sequence data of gut microbiota described in the study stored in a public database? The manuscript did not mention the database and collection number in the Data Availability Statement.

Therefore, I think that this study is not suitable for publication in this journal.

Reviewer 3 Report

The study is interesting, well written and relevant. The authors adequately declare the conflict of commercial interest. In the research, populations with hypertension risk were evaluated, assuming that non-hypertensive people respond to the consumption of barley. It would have been better to have selected people with hypertension to conduct the study and not just assume that people without hypertension are responders to barley consumption. I recommend better explaining the sample calculation to prove the hypothesis.

Some comments on specific points are below.

Line 313 - "SCFAs from the Ruminococcuceae and Lachnospiraceae families of responder-specific bacteria may reduce blood pressure" Could you explain how?

Line 315 "Second, the prevention of dysbiosis may have occurred to control innate immune system and reduce vascular inflammation and thus hypertension" Please include a citation to give support to this hipotesis.

Line 318 "SCFAs, especially propionate, that are produced from dietary fiber were reported to be one of the significant agents for inhibiting angiotensin" Could you explain how?

Line 329 "giving more weight to rare species of bacteria". What do you mean?

Line 360 "but the finding that Prevotella 9 was depleted in the intestines of the responders in this study suggests that barley may inhibit inflammation in the gastrointestinal tract." This information is merely speculative...

Line 407 "Lachnospira is a reasonable bacterium". What do you mean?

Line 444. "There are several limitations in this study". Yes, I completely agree. All participants were employees of the company that manufactures barley products and it was pointed out that they consume more barley than the general Japanese population. It may be that they consume more, or it may be that they say they consume, although they do not consume or consume less than stated. In Table 1, we can see that for the non-responder group, 27% used medication. Does the medicines influenced the result of microbiota profile?

Round 2

Reviewer 1 Report

The authors answered my questions. I have no further comments.

Reviewer 2 Report

According to the comments of editors and reviewers, the overall quality of this manuscript has been greatly improved after the author's modification. I feel that it is suitable for publication in this journal.

Reviewer 3 Report

The updates of the manuscript are satisfactory.